# Effect of Different Starches on the Rheological, Sensory and Storage Attributes of Non-fat Set Yogurt

**DOI:** 10.3390/foods9010061

**Published:** 2020-01-07

**Authors:** Ali Saleh, Abdellatif A. Mohamed, Mohammed S. Alamri, Shahzad Hussain, Akram A. Qasem, Mohamed A. Ibraheem

**Affiliations:** Department of Food Science and Nutrition, King Saud University, Riyadh 11456, Saudi Arabia; Alisa@ksu.edu.sa (A.S.); msalamri@ksu.edu.sa (M.S.A.); shussain@ksu.edu.sa (S.H.); akram@ksu.edu.sa (A.A.Q.); ibraheem@ksu.edu.sa (M.A.I.)

**Keywords:** starch, yogurt, rheology, sensory, texture

## Abstract

This study was conducted to investigate the effect of various native starches on the rheological and textural properties of non-fat set yogurt. The yogurt samples were prepared while using five types of starches (potato, sweet potato, corn, chickpea, and Turkish beans). The physical properties of the prepared yogurt were analyzed while using shear viscosity, viscoelasticity, and texture analysis. The tests were performed after 0, 7, and 15 days storage. The effect of these starches on the yogurt viscoelastic properties, texture, syneresis, and sensory evaluation were determined under optimum conditions. The results showed that adding 1% starch could significantly (*p* < 0.05) reduce syneresis and improve yogurt firmness. Starches exhibited different effect on the overall quality of the yogurt due to their origin and amylose content. Regardless of the number of storage period duration, all of the samples, including the control behaved as pseudoplastic materials (*n* < 1) with various levels of pseudoplasticity. Yogurts with corn and tuber starches had the highest consistency coefficient (k), which indicated higher viscosity. The yogurt sample with chickpea starch exhibited the highest G´, making the gel more solid like. Therefore, the influence of tuber starches (potato and sweet potato) on G´ was different when compared to corn or legume starches. The behavior of the starches changed with storage time, where some starches performed better only at the beginning of the storage period duration. Wheying-off was significantly reduced, regardless of starch type. The pH of the yogurt remained unchanged through storage. Sensory evaluation showed a preference for starch-containing samples as compared to the control, regardless of the starch type. The variation in yogurt quality as a function of starch type could be attributed to the starch granule structure, gelatinization mechanism, or amylose content.

## 1. Introduction

Nowadays, yogurt is considered to be one of the most popular fermented milk products and it has gained widespread consumer acceptance as a healthy food [1]. It is prepared by fermenting milk with bacterial cultures consisting of a mixture of *Streptococcus thermophiles* and *Lactobacillus* [2]. Yogurt is as nutritious as many other fermented milk products, since it contains high level of milk solids in addition to the nutrients developed during the fermentation process. In the yogurt industry, the major concern is the production and maintenance of a product with optimal consistency and stability during transportation and storage. The texture, viscosity at high total solids, variation in the processing variables, and characteristics of the starter culture are the main components that determine yogurt’s consistency [3]. 

The consumption of whole fat products (e.g., full fat yogurt) has declined due to the awareness of the probable harmful effect of fat on consumer’s health, thus the dietary habits of consumers have changed and market interest moved in favor of low or nonfat dairy products [4]. The Code of Federal Regulations of FDA reported that the low fat yogurt and nonfat yogurt are similar in description to full fat yogurt, but low fat contains 0.5% to 2% and nonfat is less than 0.5% milk fat [5,6].

In yogurt products, milk fat plays a major role in the texture, flavor, and color development of the final products [7]. Therefore, the reduction in fat will subsequently reduce the total solids content (in low-fat and nonfat yogurt), leading to weak body, poor texture, and increased whey separation, unless various stabilizers are used [8].

Food hydrocolloids, such as starches, are usually used in the food industries as thickeners, stabilizer, gelling agents, syneresis controller, and emulsifiers [9,10]. On the other hand, they regulate flavor and aroma release [11]. The use of stabilizers in manufactured dairy products, such as yogurt, is very important for appropriate viscosity, sensory properties, and inhibiting/reducing wheying-off during storage and transportation, as well as boosting the ratio of total solids. There are many kinds of stabilizers, for instance, synthetic (carboxyl methyl cellulose) or natural. Plant origin stabilizers are considered to be the cheapest and they include the commonly used ones, such as corn starch. Starch is preferred in the yogurt industry, because it is a good thickener and its ability to reduce yogurt flaws by improving texture and make the product more appealing to consumers [12,13,14]. Starches, such as corn, sweet potato, potato, and chestnut, are commonly used by the yogurt industry at 0.25 to 1%”. Although the use of starch in yogurt manufacturing is currently practiced, a comparison between the performance of starches form different sources and dissimilar amylose content, such as tubers, cereals, or legumes, is not done to the best of our knowledge. Therefore, the focus of this study was to explore the effect of corn, sweet potato, potato, Turkish beans, and chickpea starches on the physicochemical, rheological, and sensory properties of the none-fat set yogurt during and after storage.

## 2. Materials and Methods

### 2.1. Materials

Nonfat milk powder (34.5% protein, 3.5% moisture, 7.2% ash, 55% lactose) of Nino brand was purchased from a local supermarket. Fresh potato and sweet potato were obtained from the produce market (Riyadh, Saudi Arabia). Chickpea and Turkish beans were purchased from local store, whereas ARASCO (Riyadh, Saudi Arabia) donated the corn starch.

### 2.2. Starches Extraction

#### 2.2.1. Potato and Sweet Potato Starches Isolation

The potato or sweet potato starch was extracted according to [15]. The tuber was thoroughly washed, peeled, and diced into small pieces. Slurry was prepared by blending the diced tubers in distilled water (50:50 *v/v*) for 3 min. while using kitchen aid blender at medium speed (B. Braun Melsungen, AG, Hessen, Germany). The slurry was filtered through a muslin cloth and the overs were re-suspended in distilled water (1:2 *v*/*v*), blended, and filtered in the same way, and finally the filtrate is sieved while using 200 mesh sieve. Starch was allowed to settle for 1 h at room temperature and the supernatant was discarded. The starch was re-suspended in distilled water and then centrifuged at 2000× *g* for 15 min. After centrifugation, the top dark layer was removed and the white material at the bottom of the bottle is reconstituted in distilled water and then centrifuged until pure white starch fraction is obtained. The isolated starch is dried, ground, and stored as in air tight jars at 5 °C.

#### 2.2.2. Chickpea and Turkish Bean Starches Isolation

Whole meal of Turkish beans was prepared by crushing the dry beans in the blender at fast speed for 3 min. Slurry was prepared by blending the whole meal of chickpea or Turkish beans in distilled water (50/50; *w*/*w*) in heavy duty blender (B. Braun Melsungen, AG, Hessen, Germany) for 5 min. at medium speed. The slurry was passed through 200 mesh sieves and the filtrate was centrifuged at 2000× *g* for 15 min. [16,17]. After centrifugation, the top layer on the precipitate was removed and the white material at the bottom of the bottle (the pellet) was then re-suspended in distilled water and centrifuged while using the above-mentioned conditions. This procedure was repeated five times to get the pure white starch. The starch was air dried, ground in a coffee grinder, and stored in air-tight glass bottles at 4 °C for further analysis.

### 2.3. Amylose Content

Amylose content was determined according to the method of Williams, et al. [18]. The method is based on weighing 0.1 g of starch dry basis, 1 mL ethanol, 9 mL NaOH (I M). The mixture is boiled for 10 min. in water bath and then cooled to room temperature. To 5 mL mixture, 1 mL acetic acid (1 N), 2 mL iodine solution (2.0 g of potassium iodide and 0.2 g of iodine diluted to 100 mL with distilled water) and the absorbance (A) was red at 620 nm. The percent amylose content was calculated, as follows: 3.06 × A × 20. 

### 2.4. Yogurt Preparation

Nonfat yogurt was prepared while using powdered skim milk (14.0% total solids) and starch. The blends were prepared by replacing 1.0 g of the skim milk with 1.0 g of starch. So as to ascertain complete solubility, dry starch and milk powder were mixed first. To the dry ingredients, water was added to finally maintain 14% total solids and the mixture was preheated to 60 °C for 30 min., cooled to 42 °C, and the yogurt starter was added at 3.0% of the dry ingredients. The microbial starter included Streptococcus thermophiles and Lactobacillus bulgaricus. The mixture (14 g) was divided into plastic cups (100 mL) and incubated at 42 °C until coagulation occurs or the pH reached 4.6 (Barrantes et al., 1994) [4]. The yogurt samples were stored at 5 ± 0.2 °C and then analyzed after 0, 7, and 15 days of storage. Three replicates were tested. The pH of yogurt samples was determined at 25 °C.

### 2.5. Yogurt Composition 

#### 2.5.1. Total Solids Contents

The total solids content of yogurt samples were determined according to the Association of Official Agricultural Chemists AOAC Method (940.09) [19]. Yogurt sample (10 g) was dried in air forced oven at 105 ± 5 °C for 1 h. The remaining weight was expressed as percent total solids content. 

#### 2.5.2. Total Ash Content

The ash content of the yogurt samples was determined according to the AOAC Method (942.05) [19], where 5 g of yogurt sample was heated in a muffle furnace at 550 °C for 5 h and the residue was expressed as % Ash content. 

#### 2.5.3. Crude Protein

The crude protein content was estimated according to the Kjeldahl method, as described in the AOAC method (992.15) [19]. The yogurt sample (2 g) was digested in concentrated sulfuric acid and the total nitrogen content was multiplied by 5.70.

#### 2.5.4. Crude Fat

The crude fat content was determined while using the Gerber Method and the percent crude fat was determined by directly reading calibrated butyrometer (Badertscher et al., 2007) [20].

#### 2.5.5. Total Carbohydrates

The carbohydrate contents were determined by the difference method given using the expression given below:(1)Carbohydrates (%)=100−(protein %+fat %+moisture %+ash %).

### 2.6. Apparent Viscosity

The yogurt viscosity measurements were carried out at ambient temperature (25 °C) while using Brookfield viscometer (Brookfield Engineering Inc., Model RV-DV II Pro+, New York, NY, USA) with spindle number 63. The disk spindle was selected because of the nature of the yogurt sample, which allows for readings within the instrument sensitivity. According to the manufacturer (Brookfield), samples should be kept in a thermostatically controlled water bath at 25 °C for about 10 min. before measurements. The first measurements were taken 2 min. after the spindle was immersed in the sample to allow for thermal equilibrium to eliminate the effect of immediate time dependence effect. The data were collected every 40 sec and the measurement was duplicated and the average value was considered. The experimental data obtained was converted to shear rate/shear stress and fitted to the power law model.
σ = *k* × γ*^n^*(2)
where σ is shear stress (Pas), γ is shear rate (s^−1^), *n* is the flow behavior index, and *k* is the consistency index (Pas). The values for the flow behavior index *n* were obtained from plotting the log of shear stress versus log of shear rate and the slope of the line (if the dependence is sufficiently close to a linear one) is simply equal to the flow index (*n*).

### 2.7. Dynamic Rheology, Steady Flow Behavior

The dynamic viscoelastic properties of nonfat yogurt were determined while using TA Instrument Discovery Hybrid Rheometer (HR-1) that was installed with parallel plates system (40 mm in diameter and 50 μm gaps (TA Instruments, New Caste, PA, USA). The samples were transferred to the plate and rheometer was calibrated at 25°C for one minute and excess sample was wiped off with spatula. Dynamic shear data were obtained at frequency sweeps ranging from 0.1–100 rad/s and 0.5% constant strain at 25 °C. Experimental data were collected using the software that was provided by the manufacturer. The storage moduli (G′), loss moduli (G″), and viscosity are the parameters obtained for every run. A strain-sweep experiment was performed to establish that all measurements were done within the linear viscoelastic rage of the experiment. Linear viscoelasticity indicates that the measured parameters are independent of shear strains. To determine the liner viscoelastic range (LVR), a stress sweep was increased from 0.1 to 50.0 Pas at a constant frequency of 0.1 Hz (0.628 rad/s). Frequency sweeps between 0.1 to 10 (rad/s) were implemented within the LVR of the yogurt samples at a constant stress of 1.0 Pas. The frequency range used here is typically used for frequency sweep to ascertain that G′, G″, and η* were within the linear region. The behavior of all the measured materials in this study was in the linear range below 1% strain. Measurement was repeated three times with fresh samples. The relative errors were within the range of ±10%. All of the calculations were performed with the Rheology Advantage Data Analysis software (Version 5.7.0., TA Instruments, New Caste, PA, USA) that was provided by the manufacturer. The viscosity profile was used to provide information regarding the possibility of slippage, where the viscosity profile as a function of shear rate of duplicate runs was plotted in the same graph to ascertain the repeatability and slippage behavior of the material. No slippage was recorded. 

### 2.8. Yogurt Texture Profile Analysis (TPA)

The double compression test was performed while using texture analyzer (TA-XT2 Texture Analyzer, Texture Technologies Crop, Scarsdale, NY, USA) equipped with a software. Plastic cylinder (45 Perspex Cone, 432–081) is attached to crosshead moving at speed of 70 mm/min. in both upward and downward directions. The yogurt sample was placed on a flat holding plate and the plastic cylinder was inserted 20 mm below the surface of the yogurt sample. The firmness of the yogurt was calculated according to the method that was described by Steffe 1996 [21].

### 2.9. Whey Separation (Wheying-Off)

The volume of the separating whey (wheying-off) on the surface of the yogurt sample was collected as an indicator for wheying-off (g/100 g yogurt) according to the siphon method. The level of spontaneous whey separation of undisturbed set yogurt was determined while using a siphon drainage method [22]. In this study, a cup of set yogurt was taken out of the cold room (4 °C), weighed, and kept at an angle of approximately 45 °C to allow for whey collection on the side of the cup. A syringe was used to draw the whey from the surface of the sample every 10 s, and the cup of yogurt was weighed again. The syneresis was expressed as the percent weight of the whey over the initial weight of the yogurt samples.

### 2.10. Sensory Evaluation

The sensory evaluation was carried out by a group of 10 trained sensory assessors. The evaluation of the yogurt samples included the following sensory attributes; appearance, color, texture, aroma, taste, aftertaste, and overall acceptability. The scale used was nine points hedonic scale, where 1 represents dislike extremely, 5 for neither dislike or like, and 9 for extremely like.

### 2.11. Statistical Analysis

Statistical analysis was performed while using SPSS (SPSS Inc., Hong Kong, China). All of the measurements were done in triplicate and then subjected to analysis of variance (ANOVA) using factorial design. Duncan’s multiple range tests at *p* ≤ 0.05 was used to compare means while using PASW^®^ Statistics 18 software (SPSS Inc., Hong Kong, China).

## 3. Results and Discussion

### 3.1. Shear Viscosity 

Three different sets of temperatures were selected, 36–38 °C, 40–42 °C, and 44–46 °C, in order to determine the appropriate incubation temperature. The yogurt made at all three sets of temperature was tested for its texture, viscosity and wheying-off. Unlike the other temperatures, the yogurt that was prepared at 40–42 °C had good texture and viscosity as well as less wheying-off. Therefore, yogurt with or without starch was prepared at 42 °C. In addition, the pH was monitored throughout the incubation time and was found to decrease with the same rate in the presence of starch when compared to the control. Williams et al. (2003) [23] reported that when a modified waxy maize starch was added to yogurt prepared at 43 °C, the product had a grainy texture and high viscosity. The authors were able to improve the texture and viscosity by lowering the fermentation temperature to 35 °C and then increased the fermentation time. In this work, we used common native starch (non-waxy) and, by trying different temperatures, we found that 42 °C was the best.

The proximate composition of the yogurt showed no significant difference between the control and the yogurt/starch blends with respect to the total solids, lipids, and ash. However, lower carbohydrates and higher protein content were recorded for the control. The percent amylose content for potato, sweet potato, corn, chickpea, and Turkish bean starches were 21.8, 22.9, 20.4, 32.2, and 17.5, respectively. Figure 1 shows the effect of the starches on the shear rate and shear stress of the yogurt. By observing the shape of the curves in Figure 1, it is clear that yogurt exhibited shear thinning and yield stress behavior that indicates interactive structure [24], where the entangled molecules of the yogurt start to disentangle and become less resistant to flow which results in shear thinning as indicated by lower viscosity. Yield stress is the applied stress at which irreversible deformation is first observed across the sample. Yield stress studies can help to evaluate the product performance and process-ability and predict product’s long-term stability and shelf life. A higher yield stress prevents the material to undergo phase separation or break down and it reduces flow under shipping vibrations. At the beginning of cold storage at, the yield stress of the control and the yogurt/starch blends can be ranked as: Turkish beans > corn > chickpea > potato > sweet potato > control (Figure 1a). The gaps between the profiles (curves) in Figure 1a point to similar effect of the tubers, where the chickpea was in the middle. The highest yield stress of corn and Turkish starches indicates structural stability of the gel. After seven days of storage, the yield stress of potato starch ranked first, Turkish beans and sweet potato were the lowest and the control became firmer as compared to zero storage days (Figure 1b). The gaps between the profiles were further apart than those of at the beginning of cold storage. The sweet potato and Turkish bean were similar, whereas the control was similar to the chickpea. After 15 days, the picture was not very different than seven days, except that chickpea starch exhibited a higher value (Figure 1c). The role of these starches appeared to be time dependent, because Turkish-beans starch exhibited the highest yield stress at zero storage days, while it has the least yield stress after seven days of storage. This could be due to the mechanism of starch gelatinization during the first steps of yogurt making. Another reason could be amylose retrogradation and syneresis (water separation due to amylose retrogradation). Therefore, the use of potato or corn starch for set yogurt to maintain good gel firmness is recommended for longer storage time. Yield stress was the highest at seven storage days than zero or 15 days, which could be attributed to maximum physical interaction between the starch and the casein. Shear viscosity analysis showed Turkish-beans as starch with the highest yield stress and stability at the beginning of cold storage, whereas potato starch with the highest yield stress and stability after 15 days of storage.

The power law was applied to describe the rheological behavior of the yogurt with different types of starches (Equation (1)). Table 1 a,b present the effect of the starches on the power law parameters (*k* and *n*), where the data clearly fit the power law, because of the values of the high value of *r*^2^ (>0.95). The Y-intercept represents the K and the slope represents the *n*. Regardless of the number of storage days, all the samples (Table 1), including the control behaved as pseudoplastic materials, since *n* < 1, but some are more pseudoplastic than others. Cruz et al. (2012) [25]; Yu et al. (2007) [26] noticed a similar observation. The difference in pseudoplasticity of yogurts with different types of starch as a function of storage time is obvious in Table 1. This is an indication of the structural changes during storage due to interactions between starch molecules, possibly amylose, and the casein network. Unlike the control and the corn starch samples, and by virtue of higher *n* values, yogurt sample containing Turkish beans starch was the least pseudoplastic, especially at longer storage time. This behavior was also reflected for the yield stress. Potato and sweet potato starches exhibited the most pseudoplasticity after seven days storage. This difference can be attributed to the different amylose content of these starches, where the lowest amount of amylose showed the smallest pseudoplasticity. Conversely, chickpea with the highest amylose content started as the least pseudoplastic, but it become more pseudoplastic as a function of storage time (Table 1). When compared to the control, all of the yogurt/starch blends exhibited higher *k* values, which indicated higher viscosity and a thicker structure (Table 1). The k values at the beginning of cold storage was similar for all starches, regardless of amylose content, but for longer storage time corn starch yogurt exhibited the thickest texture followed by the tubers. Once again, the Turkish bean starch yogurt was the thinnest of all samples, which could be accredited to the low amylose content. The thickest yogurt after 15 days storage was the one with potato starch, which was also true for the yield stress results. Other researchers attributed the high yogurt viscosity to the high milk solids [27].

### 3.2. Viscoelastic Properties

The dynamic rheological testing provides useful information regarding the internal structure of the yogurt, which is represented by storage modulus (G′) and loss modulus (G″) that denote the elastic and the viscous behavior, respectively, as well as the complex viscosity η* and the phase angle tan δ. The yogurt samples exhibited a weak gel behavior because G′ > G″ over the range of 0.1 and 50 Pas. A linear viscoelastic region was observed from 0.1 to 10 Pas. Therefore, 1.0 Pas stress was chosen for the frequency sweep test. These data are in agreement of other researchers who reported similar results for yogurt enriched with milk solids and inulin [28]. The longer LVR G′ region of the yogurt samples as a function of stress indicates stress independence (Figure 2). The extent of LVR of the yogurt samples can be ranked according to the type of starch, as follows: potato > corn > chickpea > sweet potato > Turkish beans > control (Figure 2). Therefore, the gel of the control yogurt was the weakest of all and potato starch produced yogurt with the firmest gel (structure), whereas yogurt with Turkish beans starch exhibited the least firm gel. These data are in agreement with the yield stress and the *k* and *n* values reported above.

Figure 3 showed the oscillation dependency of G′. The G′ of the control stayed the same up to seven storage days, but it increased after that. Chickpea starch was more effective than other starches by maintaining the highest G′; nonetheless, G′ was almost the same after seven or 15 days (Figure 3b). Although on smaller scale, the G′ corn starch yogurt was similar to chickpea starch (Figure 3c), where the gel became firmer as a function of longer storage time. The firmest gel for potato and sweet potato was recorded after seven storage days i.e., high G′, but potato starch yogurt was softer after 15 days (Figure 3e,d). Generally, the G′ of sweet potato starch was higher than potato starch which can be attributed to the higher amylose content of the sweet potato starch. The remaining starches showed higher G′ after 15 days. In addition, the G′ appeared to be unchanged after seven and 15 days, except for the tuber starches (Figure 3b,c). Reports indicated that chemically modified starch can induce positive impact in syneresis and rheological properties as compared with a full-fat yogurt by forming a stable yogurt structure [13]. The texture of low fat yogurt that was prepared with acid treated crosslinked wheat starch was soft, but syneresis was lower [29]. Acetylated-crosslinked starch improved the properties of yogurt more effectively than native starch at 0.5% concentration, in terms of yield stress, consistency, apparent viscosity, thixotropy, and pseudoplasticity [30]. Generally, yogurt with chickpea starch exhibited the highest G′ after 15 days in storage (Figure 3f). Overall, the G′ of the tubers was similar and the legumes starch yogurt were alike, because the yogurt gels became firmer with time. In the presence of legume starches, a large G′ gap between zero and seven days storage was observed when compared to small G′ gap between the same storage days of the tubers. The G′ of the tested yogurts appeared to reach its maximum at seven days storage and remained unchanged, except for tuber starches.

### 3.3. Yogurt Texture Profile Analysis

Generally, yogurt texture depends on the physical interaction between casein micelles [31]. Overall, yogurt that was prepared with or without starch exhibited harder texture with storage time, regardless of starch type (Table 2 a–c). Corn and sweet potato starches increased the hardness of the yogurt significantly (*p* < 0.05) after seven days, but the effect of corn starch dropped after 15 days. This could be attributed to weaker corn starch amylose network, which allows for the water to be free after it has been trapped in the gel network. Yogurt containing tuber starches (sweet potato and potato) exhibited the hardest gel through most of the storage time (Table 2). The highest yogurt hardness was recorded for chickpea starch after 15 storage days and corn starch after zero and seven days. Sweet potato starch was the only starch performed well at the three storage times followed by potato and corn. Chickpea starch appeared to significantly increase hardness after 15 days relative to the other starches (Table 2 c), which is in line with the highest G′ for chickpea starch mentioned earlier. This could be accredited to the high amylose content capable of forming stronger network. The cohesiveness is the degree of yogurt deformation during testing. Turkish beans yogurt exhibited the highest cohesiveness, which points to softer gel, as pointed by the hardness data. Adhesiveness is the attractive force between the food and the teeth that can predict the stickiness of the food. The stickiest yogurt is the one with chickpea starch and the control (Table 2). Generally, the higher value of adhesiveness suggests softer yogurt texture; this was not true for the Turkish beans starch, where soft gel faces high adhesiveness (Table 2), but this starch exhibited the highest cohesiveness value of all the starches. Once again, Turkish beans behaved differently from other starches. Gumminess is defined as the product of hardness and cohesiveness that is typical of semisolid foods with a low degree of hardness and a high degree of cohesiveness. Chickpea starch had the highest gumminess value in seven days and after which can be attributed to the high hardness. Texture analysis showed chickpea starch with the hardest yogurt through the storage period, whereas the tubers exhibited the highest hardness until seven days of storage. Therefore, it is recommended to use chickpea starch for longer storage time. 

### 3.4. Whey Separation

Wheying-off is a negative characteristic of set yogurt and it is defined as the expulsion of whey from the casein network. Spontaneous wheying-off is the separation of whey without the application of any external force that is associated with unstable gel network. This can be caused by increased rearrangements of the gel matrix or by mechanical damage to the weak gel network. Manufacturers used stabilizers, such as starch, pectin, and gelatin, to prevent wheying-off [11,32,33]. Common causes of wheying-off include high incubation time, disproportionate whey protein to casein ratio, low solid content, and physical mishandling of the product during storage and distribution. Yogurt treated with 1% crosslinked cassava, corn starch, and tapioca starch significantly reduced yogurt syneresis [32]. Altemimi 2018 [33] reported that the reduction in wheying-off by potato starch was significant. The reported data showed no significant difference between 0.25–0.50% potato starch and the control, but 0.75 and 1.00% significantly reduced wheying-off [33]. The results of this work showed a significant reduction (*p* < 0.05) in wheying-off, regardless of starch type for the same storage time (Figure 4). As a function of storage time, wheying-off significantly increased after seven or 15 days, but the rise was significantly less than that of the control. Therefore, wheying-off was more affected during storage and not by starch type. Significant syneresis reduction can be achieved by using modified starches when compared to native starches, as reported by other researchers [34]. With respect to wheying-off, all of the starches significantly reduced wheying-off the same way without any preference. Hence, starch selection should be based on other yogurt attributes, where starch performance was significantly different. 

### 3.5. Sensory Evaluation

Yogurt texture is one of the main characteristics that define its sensory quality and affect appearance, mouth-feel, and overall consumer acceptability. In addition, yogurt consistency is perhaps as important as flavor. Acceptable firmness without syneresis is critical for excellent final product. Yogurt texture is usually measured in the cup by using a spoon or directly in the mouth, where the viscosity of the product can represent yogurt consistency on the tongue. The yogurt sample is considered to be thick (viscous) if it stays on the tongue or flows slowly and is swallowed with difficulty, while the visually thick sample can be measured by tilting the spoon and observing how slow the sample will flow. The statistical analysis of this work underlined that there was no significant difference (*p* > 0.05) between the starches with respect to sensory texture, but it was significantly better than the control (Table 3). The sensory viscosity of sweet potato and corn starches were significantly higher than the control, where other starches exhibited similar values. The best flavor was recorded for yogurt with corn starch and chickpea starch, whereas all the starches showed better flavor than the control. The panelist overall acceptability was not significantly different, but the starch containing samples were better accepted, especially yogurt with sweet potato starch. The panel preferred yogurt that was made with sweet potato starch the most.

## 4. Conclusions

Stabilizers are important ingredients in manufactured yogurt or other dairy products due to their capacity to improve viscosity and sensory properties, and to decrease wheying-off during storage. The results showed that adding 1% starch gave better results than the control for the syneresis, sensory evaluation, and viscoelastic properties. The outcome of this work showed that tuber starches can behave differently from cereal or legume starches. In addition, the data indicated that the storage time of yogurt could affect starches behavior. The variation in the effects of starches on yogurt quality could be due to the starch granule structure, gelatinization mechanism, and amylose content. The overall results obtained indicate tuber starches or chickpea starch as the most promising stabilizer of non-fat yogurt during storage. 

## Figures and Tables

**Figure 1 foods-09-00061-f001:**
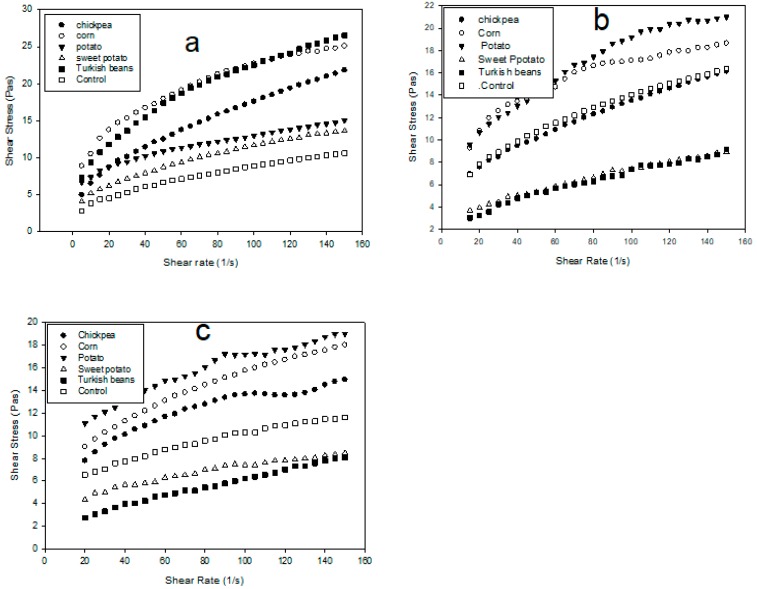
Shear rate shear stress relationship for fresh yoghurt fortified with deferent starches after (**a**) 0 storage days, (**b**) 7 storage days, and (**c**) 15 storage days.

**Figure 2 foods-09-00061-f002:**
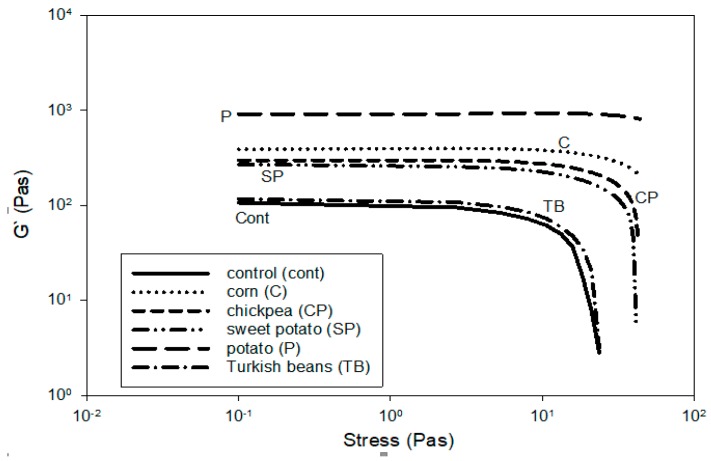
Linear viscoelastic region determination of the control and starch containing yogurt.

**Figure 3 foods-09-00061-f003:**
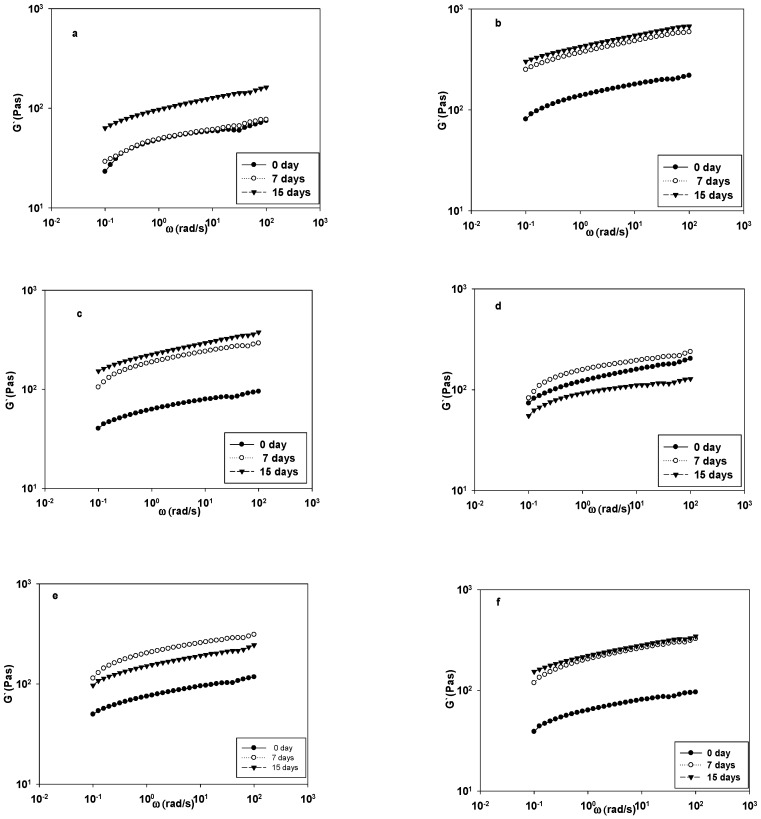
The G′ of yoghurt samples fortified with starch and stored for 0, 7, and 15 days: (**a**) control, (**b**) chickpea, (**c**) corn, (**d**) potato, (**e**) sweet potato, and (**f**) Turkish beans starches.

**Figure 4 foods-09-00061-f004:**
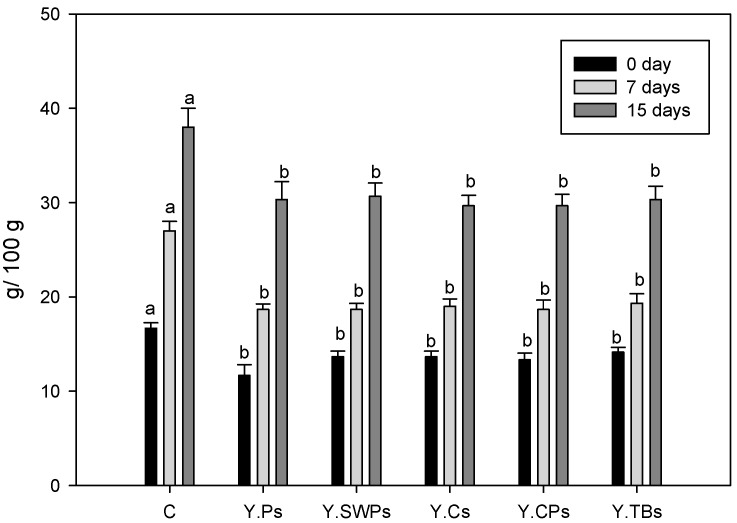
Effect of starches addition on whey separation of yogurt. C = Control, Y. Ps = potato starch, Y. SWPs = Sweet potato starch, Y. Cs = corn starch, Y. CPs = Chickpea starch, Y. TBs = Turkish bean starch Yogurt samples with the different types of starch with the same letter are not significantly different.

**Table 1 foods-09-00061-t001:** The consistency index (*k*) and flow behavior index (*n*) of yogurt prepared with different starches.

**1a**	**0 Day**		**7 Days**		**15 Days**	
	***k***	***r^2^***	***k***	***r*^2^**	***k***	***r*^2^**
Control	0.381 ± 0.01 ^b^	0.991 ± 0.02	0.401 ± 0.08 ^c^	0.992 ± 0.02	0.471 ± 0.06 ^c^	0.991 ± 0.01
Y. Ps	0.423 ± 0.02 ^a^	0.989 ± 0.02	0.542 ± 0.02 ^b^	0.994 ± 0.00	0.681 ± 0.01 ^a^	0.992 ± 0.01
Y. SWPs	0.423 ± 0.03 ^a^	0.993 ± 0.01	0.550 ± 0.05 ^b^	0.989 ± 0.02	0.235 ± 0.05 ^e^	0.995 ± 0.01
Y. Cs	0.423 ± 0.03 ^a^	0.998 ± 0.02	0.682 ± 0.02 ^a^	0.979 ± 0.02	0.496 ± 0.03 ^b^	0.990 ± 0.00
Y. CPs	0.424 ± 0.06 ^a^	0.996 ± 0.03	0.381 ± 0.03 ^c^	0.990 ± 0.05	0.324 ± 0.02 ^d^	0.975 ± 0.03
Y. TBs	0.431 ± 0.08 ^a^	0.997 ± 0.07	0.271 ± 0.01 ^d^	0.995 ± 0.03	0.101 ± 0.021 ^f^	0.990 ± 0.02
**1b**	**0 Day**		**7 Days**		**15 Days**	
	***n***	***r^2^***	***n***	***r*^2^**	***n***	***r*^2^**
Control	0.381 ± 0.11 ^b,c^	0.991 ± 0.02	0.310 ± 0.01 ^d^	0.992 ± 0.02	0.294 ± 0.04 ^d^	0.991 ± 0.01
Y. Ps	0.246 ± 0.02 ^c^	0.989 ± 0.02	0.365 ± 0.01 ^c^	0.994 ± 0.01	0.275 ± 0.05 ^e^	0.992 ± 0.01
Y. SWPs	0.370 ± 0.11 ^c^	0.993 ± 0.01	0.410 ± 0.03 ^b^	0.989 ± 0.02	0.317 ± 0.11 ^c^	0.995 ± 0.01
Y. Cs	0.313 ± 0.03 ^d^	0.998 ± 0.02	0.280 ± 0.02 ^e^	0.979 ± 0.02	0.349 ± 0.22 ^b^	0.990 ± 0.02
Y. CPs	0.440 ± 0.01 ^a^	0.996 ± 0.03	0.380 ± 0.03 ^c^	0.990 ± 0.05	0.304 ± 0.42 ^d^	0.975 ± 0.03
Y. TBs	0.390 ± 0.01 ^b^	0.997 ± 0.07	0.488 ± 0.22 ^a^	0.995 ± 0.03	0.535 ± 0.23 ^a^	0.990 ± 0.02

*k* consistency index (Pa). *n* = flow behavior index (dimensionless). C = Control, Y. Ps = potato starch, Y. SWPs = Sweet potato starch, Y. Cs = corn starch, Y. CPs = Chickpea starch, Y. TBs = Turkish bean starch. Values followed by different letters in each column are significantly different (*p* ≤ 0.05).

**Table 2 foods-09-00061-t002:** Effect of starches addition on texture of yogurt after 0, 7, and 15 days of storage.

	Hardness (g)	Cohesiveness	Adhesiveness (mJ)	Gumminess (g)
2a	0 days storage			
Control	21.000 ± 1.500 ^d^	0.351 ± 0.021 ^c^	0.471 ± 0.110 ^a^	7.351 ± 0.560 ^c^
Y. Ps	22.230 ± 0.581 ^c,d^	0.382 ± 0.011 ^b^	0.300 ± 0.000 ^b^	8.561 ± 0.200 ^b^
Y. SWPs	27.670 ± 0.580 ^b^	0.272 ± 0.022 ^e^	0.330 ± 0.060 ^b^	7.471 ± 0.300 ^c^
Y. Cs	30.330 ± 0.588 ^a^	0.321 ± 0.011 ^d^	0.332 ± 0.060 ^b^	9.810 ± 0.200 ^a^
Y. CPs	22.670 ± 0.580 ^c^	0.323 ± 0.011 ^d^	0.532 ± 0.060 ^a^	7.331 ± 0.060 ^c^
Y. TBs	15.330 ± 1.530 ^e^	0.433 ± 0.011 ^a^	0.302 ± 0.002 ^b^	6.541 ± 0.510 ^d^
2b	7 days storage			
Control	24.000 ± 1.111 ^d^	0.461 ± 0.011 ^a^	0.330 ± 0.060 ^a^	10.660 ± 0.500 ^c^
Y. Ps	29.000 ± 1.000 ^b^	0.212 ± 0.020 ^d^	0.130 ± 0.062 ^b^	6.190 ± 0.300 ^e^
Y. SWPs	29.331 ± 0.581 ^b^	0.320 ± 0.010 ^c^	0.171 ± 0.061 ^b^	9.391 ± 0.461 ^d^
Y. Cs	31.330 ± 0.580 ^a^	0.350 ± 0.010 ^b^	0.172 ± 0.120 ^b^	10.971 ± 0.51 ^c^
Y. CPs	29.272 ± 0.462 ^b^	0.471 ± 0.010 ^a^	0.300 ± 0.000 ^a^	13.702 ± 0.152 ^a^
Y. TBs	25.330 ± 0.582 ^c^	0.472 ± 0.012 ^a^	0.330 ± 0.060 ^a^	11.992 ± 0.22 ^b^
2c	15 days storage			
Control	28.000 ± 1.000 ^c^	0.420 ± 0.030 ^b^	0.571 ± 0.061 ^a^	11.850 ± 0.300 ^b^
Y. Ps	30.000 ± 1.000 ^a,b,c^	0.340 ± 0.010 ^c^	0.220 ± 0.251 ^e^	10.300 ± 0.211 ^c^
Y. SWPs	30.331 ± 0.581 ^a,b^	0.410 ± 0.010 ^b^	0.230 ± 0.060 ^d,e^	12.441 ± 0.150 ^b^
Y. Cs	28.672 ± 0.582 ^b,c^	0.360 ± 0.010 ^c^	0.301 ± 0.001 ^c^	10.222 ± 0.372 ^c^
Y. CPs	32.232 ± 2.023 ^a^	0.420 ± 0.010 ^b^	0.431 ± 0.061 ^b^	13.552 ± 1.222 ^a^
Y. TBs	21.000 ± 1.111 ^d^	0.460 ± 0.010 ^a^	0.231 ± 0.061 ^d,e^	9.661 ± 0.600 ^c^

Mean of three replicates. Values followed by different letters in each column are significantly different (*p* ≤ 0.05). C = Control, Y. Ps = potato starch, Y. SWPs = Sweet potato starch, Y. Cs = corn starch, Y. CPs = Chickpea starch, Y. TBs = Turkish bean starch.

**Table 3 foods-09-00061-t003:** Effect of starches on the sensory evaluation of yogurt.

Treatments	Viscosity	Texture	Creaminess	Flavor	Overall Acceptability
Control	5.40 ± 1.07 ^b^	5.11 ± 1.50 ^b^	6.30 ± 1.34 ^a,b^	6.80 ± 1.39 ^b^	5.90 ± 0.99 ^a,b^
Y. Ps	6.40 ± 1.18 ^a,b^	6.21 ± 1.05 ^a^	6.51 ± 0.82 ^a^	7.50 ± 0.53 ^a,b^	6.30 ± 0.67 ^a,b^
Y. SWPs	6.50 ± 1.34 ^a^	6.33 ± 0.99 ^a^	6.40 ± 1.08 ^a,b^	7.70 ± 1.05 ^a,b^	6.50 ± 1.08 ^a^
Y. Cs	6.70 ± 0.84 ^a^	6.13 ± 0.63 ^a^	5.91 ± 1.07 ^a,b^	7.90 ± 0.32 ^a^	6.10 ± 1.10 ^a,b^
Y. CPs	6.40 ± 0.84 ^a,b^	6.23 ± 0.63 ^a^	6.12 ± 0.86 ^a,b^	8.10 ± 1.07 ^a^	6.20 ± 0.92 ^a,b^
Y. TBs	6.40 ± 0.83 ^a,b^	6.23 ± 0.63 ^a^	6.14 ± 0.84 ^a,b^	7.40 ± 0.60 ^a,b^	6.21 ± 0.92 ^a,b^

C = Control, Y. Ps = potato starch, Y. SWPs = Sweet potato starch, Y. Cs = corn starch, Y. CPs = Chickpea starch, Y. TBs = Turkish bean starch. The scale used was 1 represents dislike extremely, 5 for neither dislike or like and 9 for like extremely. Mean of three replicates. Values followed by different letters in each column are significantly different (*p* ≤ 0.05).

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
