# Peer review of "Effect of Different Starches on the Rheological, Sensory and Storage Attributes of Non-fat Set Yogurt"

_foods, 2020, doi:10.3390/foods9010061_

Round 1
Reviewer 1 Report
The introduction needs to be improved by reviewing the existing studies and summarizing the knowledge obtained and the gaps up to date, and then assumption should be made to introduce this study. The methodology was well-described. So many analyses were done in this study. The figures and tables can be improved by highlighting what kind of differences you intended to show up. This manuscript is informative, but the writing is rough, and language needs to be corrected by a native speaker.
Line 17: “The two yogurts with corn and tuber starches had…”, one corn and two tuber starches, so there should be 3 samples;
Line 19-20: “Therefore, the influence of tuber starches (potato and sweet potato) was different compared 19 to corn or legume starches.” Contradiction to previous statement. You just said corn and tuber starches had the highest.....i expect no difference between corn and 2 tuber starches.
Line 45-48: This part needs to be largely improved. Starch has various functions, just provide relative citations related to your study (providing enough background and current knowledge gaps by referring major studies). Don’t mention dietary fiber, lower cholesterol, blood pressure, flavor and aroma etc.
Line 52-54: this part is important. Please concisely provide type of starch, concentration of starch, and the effects on yogurt quality in these cited studies;
Line 79: how was the whole meal prepared? What was the blending speed?
Line 94: “milk with 1.0 g of starch of 14.0 g”, cannot be understood;
Line 215: Fig.1 is not visible;
Line 238-243: please combine Table 1a and 1b, delete the redundant R square. You can just switch row and column to make the table fitting to the page portrait orientation or make this table in landscape orientation.
Line 274: Figure 3: the curve symbols and legends are too small to be readable;
Line 300: Table 2: two digits after the digit point should be enough. Furthermore, replace 2a by 0 day storage, 2b by 7 days storage, and 2c by 15 days storage;
Line 340: Table 3, one digit after the digit point should be ok.
Author Response
Response to Reviewer 1
At first I would like to thank you for taking the time to look at our work and we appreciated your comments
Line 17: “The two yogurts with corn and tuber starches had…”, one corn and two tuber starches, so there should be 3 samples;
Response:
The sentence is modified to “Yogurts with corn and tuber starches had the highest consistency coefficient (k) indicating higher viscosity.
Line 19-20: “Therefore, the influence of tuber starches (potato and sweet potato) was different compared to corn or legume starches.” Contradiction to previous statement. You just said corn and tuber starches had the highest.....i expect no difference between corn and 2 tuber starches.
Response:
The sentence is changed to “Therefore, the influence of tuber starches (potato and sweet potato) on G` was different when compared to corn or legume starches.
Line 45-48: This part needs to be largely improved. Starch has various functions, just provide relative citations related to your study (providing enough background and current knowledge gaps by referring major studies). Don’t mention dietary fiber, lower cholesterol, blood pressure, flavor and aroma etc.
Response:
This sentence was removed and references were added “However, their function as dietary fiber is expected to lower cholesterol and blood pressure”.
Line 52-54: this part is important. Please concisely provide type of starch, concentration of starch, and the effects on yogurt quality in these cited studies;
Response:
The following sentence was added “Starches such as corn, sweet potato, potato, and chestnut are commonly used by the yogurt industry at 0.25 to 1%”.
Line 79: how was the whole meal prepared? What was the blending speed?
Response:
This first section of Tukish beans starch isolation was changed to. “Whole meal of Turkish beans was prepared by crushing the dry beans in the blender at fast speed for 3 min. Slurry was prepared by blending the whole meal of chickpea or Turkish beans in distilled water (50/50; w/w) in heavy duty blender (B. Braun Melsungen, AG, Hessen, Germany) for 5 min at medium speed”.
Line 94: “milk with 1.0 g of starch of 14.0 g”, cannot be understood;
Response:
1.0 g of skim milk was replaced with 1.0 g of starch
Line 215: Fig.1 is not visible;
Response:
Fig1 was changed
Line 238-243: please combine Table 1a and 1b, delete the redundant R square. You can just switch row and column to make the table fitting to the page portrait orientation or make this table in landscape orientation.
Response:
Table 1a and 1b were combined
Line 274: Figure 3: the curve symbols and legends are too small to be readable;
Response:
Fig 3 was enlarged
Line 300: Table 2: two digits after the digit point should be enough. Furthermore, replace 2a by 0 day storage, 2b by 7 days storage, and 2c by 15 days storage;
Response:
Table 2 was adjusted
Line 340: Table 3, one digit after the digit point should be ok.
Response:
Was done

Reviewer 2 Report
Dear authors,
Manuscript 660253
Title: Effect of Different Starches on the Rheological, Sensory and Storage Attributes of Set Yogurt
Major comment:
The manuscript titled “Effect of Different Starches on the Rheological, Sensory and Storage Attributes of Set Yogurt” describes the effect of the addition of different starches on rheological and sensory properties of non-fat yogurt during cold storage. The comparison among the behaviour of different native starches during storage is interesting. Indeed, the literature reports the effect of modified starch (e.g. acetylated) or resistant starch addition in yogurt during storage. However, some parts not clear need to be revised.
Minor comments:
Title: the title could be changed in “Effect of Different Starches on the Rheological, Sensory and Storage Attributes of Non-fat Set Yogurt”.
L13-14 Not clear. Please rewrite the sentence.
L15-16 number of storage period? Probably storage period duration…Here and throughout the manuscript please use “storage period duration”.
L23 A conclusion should be added.
L28-30 Please revises starter taxonomy: Streptococcus thermophilus and Lactobacillus delbrueckii subsp. bulgaricus.
L88 NaOH (1 M).
L94 Please reports the percentage of starch on the total solids content.
L97 Please includes the microbial composition of the starter used and the percentage added to yogurt.
L98 accrued. Please verify this term.
L101 This sentence is not necessary. Please delete
L115-117 Please include a reference for this method.
L121. Ok. How fiber content was determined? Where is reported the related method?
L164 Please includes the year of publication.
L191-194 It is not clear. Please rewrite the sentence.
L198 Here and throughout the manuscript please replace zero days with “at the begin of cold storage”.
L213 different starches…
L223 Please includes the authors and year of publication.
L229 least? Probably “starch with lowest amylose content determined smallest pseudoplasticity”.
L313 Please includes the year of publication.
In order to allow readers to compare the behaviour of different starches, I suggest to include a conclusion statement at the end of 3.1, 3.2, 3.3, and 3.4 sections to report the main results (e.g. 3.1 Shear viscosity analysis showed Turkish-beans as starch with highest yield stress and stability at the begin of cold storage, whereas potato and corn starches showed the highest values after 15 days of storage and so on for other sections…).
Please includes statistics in Table 1 a and b, and Figure 4 with letters indicating significant differences at each sampling time.
L351 can be affected…
L353 Please includes a conclusion statement (e.g. overall the results obtained indicates tuber starches or chickpea starch as the most promising stabilizers of non-fat yogurt during storage) and a perspective (e.g. the comparison of different resistant starches or modified starches addition on rheological and sensory properties of yogurt during storage).
Why the authors did not evaluate viability of starter cultures in different samples during storage? The addition of starch could promote or negatively affect bacterial growth, or could modify the growth kinetics during cold storage. Please explain the lack of microbiological evaluations. Moreover, as previously reported (Williams, R. P. W., Glagovskaia, O., & Augustin, M. A., 2003. Properties of stirred yogurts with added starch: effects of alterations in fermentation conditions. Australian Journal of Dairy Technology, 58(3), 228.), the lowering in fermentation temperature in starch-added yogurt could positively affect rheological and sensory properties. Why authors did not evaluate this parameter to compare the behaviour of different starches?
The discussion, throughout the manuscript, need to be improved. Some papers could be considered to compare different results obtained. The following papers (but not limited to) are suggested: doi.org/10.1016/j.jfoodeng.2014.01.019; doi.org/10.3923/jas.2009.2194.2197; doi.org/10.1016/j.jfoodeng.2018.10.003; doi.org/10.3168/jds.2018-15562).
Author Response
Response to Reviewer 2
At the start, my colleagues and I would like to thank you for your valuable comments
Title: the title could be changed in “Effect of Different Starches on the Rheological, Sensory and Storage Attributes of Non-fat Set Yogurt”.
Response:
Title was changed
L13-14 Not clear. Please rewrite the sentence.
Response:
The sentence is changed to “The results showed that adding 1% starch could significantly (p<0.05) reduce syneresis and improve yogurt firmness”.
L15-16 number of storage period? Probably storage period duration…Here and throughout the manuscript please use “storage period duration”.
Response:
“storage period duration” was through out
L23 A conclusion should be added.
Response:
This conclusion was added “The variation in yogurt quality as a function of starch type could be attributed to the starch granule structure, gelatinization mechanism, or amylose content”.
L28-30 Please revises starter taxonomy: Streptococcus thermophilus and Lactobacillus delbrueckii subsp. bulgaricus.
Response:
Bacterial names are like this now” Streptococcus thermophiles and Lactobacillus Bulgaricus
”
L88 NaOH (1 M).
Response:
The sentence is as follows ”The method is based on weighing 0.1 g of starch dry basis, 1 ml ethanol, 9 ml NaOH (I M)”.
L94 Please reports the percentage of starch on the total solids content.
Response:
Starch content is. “The blends were prepared by replacing 1.0 g of the skim milk with 1.0 g of starch”.
L97 Please includes the microbial composition of the starter used and the percentage added to yogurt.
Response:
Streptococcus thermophiles and Lactobacillus Bulgaricus
L98 accrued. Please verify this term.
Response:
It should be occurred
L101 This sentence is not necessary. Please delete
Response:
The sentence was removed
L115-117 Please include a reference for this method.
Response:
This reference was added “(Badertscher et al 2007).”
L121. Ok. How fiber content was determined? Where is reported the related method?
Response:
Equation was corrected because the fiber was not part of it
L164 Please includes the year of publication.
Response:
Year was added
L191-194 It is not clear. Please rewrite the sentence.
Sentence was rewritten as “However, lower carbohydrates and higher protein content were recorded for the control”
L198 Here and throughout the manuscript please replace zero days with “at the begin of cold storage”.
Response:
It is done
L213 different starches…
Response:
Done
L223 Please includes the authors and year of publication.
Response:
Year was added
L229 least? Probably “starch with lowest amylose content determined smallest pseudoplasticity”.
Response:
Least was replaced with lowest was
L313 Please includes the year of publication.
Response:
Done
In order to allow readers to compare the behaviour of different starches, I suggest to include a conclusion statement at the end of 3.1, 3.2, 3.3, and 3.4 sections to report the main results (e.g. 3.1 Shear viscosity analysis showed Turkish-beans as starch with highest yield stress and stability at the begin of cold storage, whereas potato and corn starches showed the highest values after 15 days of storage and so on for other sections…).
Response:
Each section included a form of conclusion
Please includes statistics in Table 1 a and b, and Figure 4 with letters indicating significant differences at each sampling time.
Response:
Letters were added in Table 1
L351 can be affected…
Response:
Affect replaced effect
L353 Please includes a conclusion statement (e.g. overall the results obtained indicates tuber starches or chickpea starch as the most promising stabilizers of non-fat yogurt during storage) and a perspective (e.g. the comparison of different resistant starches or modified starches addition on rheological and sensory properties of yogurt during storage).
Response:
Each section included a form of conclusion
Why the authors did not evaluate viability of starter cultures in different samples during storage? The addition of starch could promote or negatively affect bacterial growth, or could modify the growth kinetics during cold storage. Please explain the lack of microbiological evaluations. Moreover, as previously reported (Williams, R. P. W., Glagovskaia, O., & Augustin, M. A., 2003. Properties of stirred yogurts with added starch: effects of alterations in fermentation conditions. Australian Journal of Dairy Technology, 58(3), 228.), the lowering in fermentation temperature in starch-added yogurt could positively affect rheological and sensory properties. Why authors did not evaluate this parameter to compare the behaviour of different starches?
Response:
This section was added at the beginning of the shear rate discussion” Shear viscosity
In order to determine the appropriate incubation temperature, three different sets of temperatures were selected, 36-38°C, 40-42°C, and 44-46°C. The yogurt made at all three sets of temperature was tested for its texture, viscosity and wheying-off. Unlike the other temperatures, the yogurt prepared at 40-42°C had good texture and viscosity as well as less wheying-off. Therefore, yogurt with or without starch was prepared at 42°C. In addition, the pH was monitored throughout the incubation time and found to decrease with the same rate in the presence of starch compared to the control. Williams et al (2003) [21] reported that when a modified waxy maize starch was added to yogurt prepared at 43°C, the product had a grainy texture and high viscosity. The authors were able to improve the texture and viscosity by lowering the fermentation temperature to 35°C and increased the fermentation time. In this work we used common native starch (non-waxy) and by trying different temperatures we found that 42°C was the best.”
The discussion, throughout the manuscript, need to be improved. Some papers could be considered to compare different results obtained. The following papers (but not limited to) are suggested: doi.org/10.1016/j.jfoodeng.2014.01.019; doi.org/10.3923/jas.2009.2194.2197; doi.org/10.1016/j.jfoodeng.2018.10.003; doi.org/10.3168/jds.2018-15562).
Response:
The discussion was re-checked and the suggested papers were included
[21] Williams, Roderick & Glagovskaia, O. & Augustin, Mary Ann. (2003). Properties of stirred yogurts with added starch: Effects of alterations in fermentation conditions. Australian Journal of Dairy Technology. 58. 228-232.
[27] Alakali J.S., Okonkwo T.M., Iordye E.M. Effect of stabilizers on the physico-chemical and sensory attributes of thermized yoghurt. Afr. J. Biotechnol. 2008;7:158–163.
[28] Sameen A., Sattar M.U., Javid A., Ayub A., Khan M.I. Quality evaluation of yoghurt stabilized with sweet potato (Ipomoea batatas) and taro (Colocassia esculenta) starch. Int. J. Food Allied Sci. 2016; 2: 23–29.
[29] Malik A.H., Anjum F.M., Sameen A., Khan M.I., Sohaib M. Extraction of starch from Water Chestnut (Trapa bispinosa Roxb) and its application in yoghurt as a stabilizer. Pak. J. Food Sci. 2012; 22: 209–218
